# Characterization of *Entamoeba* fatty acid elongases; validation as targets and provision of promising leads for new drugs against amebiasis

Fumika Mi-ichi[1,2]*, Hiroshi Tsugawa[3,4,5], Tam Kha Vo[1,2], Yuto Kurizaki[3], Hiroki Yoshida[2], Makoto Arita[4,5,6,7]

1 Central Laboratory, Institute of Tropical Medicine (NEKKEN), Nagasaki University, Nagasaki, Japan, 2 Division of Molecular and Cellular Immunoscience, Department of Biomolecular Sciences, Faculty of Medicine, Saga University, Saga, Japan, 3 Department of Biotechnology and Life Science, Tokyo University of Agriculture and Technology, Tokyo, Japan, 4 Laboratory for Metabolomics, RIKEN Center for Integrative Medical Sciences, Yokohama, Japan, 5 Graduate School of Medical Life Science, Yokohama City University, Yokohama, Japan, 6 Division of Physiological Chemistry and Metabolism, Graduate School of Pharmaceutical Sciences, Keio University, Tokyo, Japan, 7 Human Biology-Microbiome-Quantum Research Center (WPI-Bio2Q), Keio University, Tokyo, Japan

* fumika@nagasaki-u.ac.jp

**Data Availability Statement:** The data that support the findings of this study are publicly available at

## Abstract

*Entamoeba histolytica* is a protozoan parasite belonging to the phylum *Amoebozoa* that causes amebiasis, a global public health problem. *E. histolytica* alternates its form between a proliferative trophozoite and a dormant cyst. Trophozoite proliferation is closely associated with amebiasis symptoms and pathogenesis whereas cysts transmit the disease. Drugs are available for clinical use; however, they have issues of adverse effects and dual targeting of disease symptoms and transmission remains to be improved. Development of new drugs is therefore urgently needed. An untargeted lipidomics analysis recently revealed structural uniqueness of the *Entamoeba* lipidome at different stages of the parasite's life cycle involving very long (26–30 carbons) and/or medium (8–12 carbons) acyl chains linked to glycerophospholipids and sphingolipids. Here, we investigated the physiology of this unique acyl chain diversity in *Entamoeba*, a non-photosynthetic protist. We characterized *E. histolytica* fatty acid elongases (EhFAEs), which are typically components of the fatty acid elongation cycle of photosynthetic protists and plants. An approach combining genetics and lipidomics revealed that EhFAEs are involved in the production of medium and very long acyl chains in *E. histolytica*. This approach also showed that the K3 group herbicides, flufenacet, cafenstrole, and fenoxasulfone, inhibited the production of very long acyl chains, thereby impairing *Entamoeba* trophozoite proliferation and cyst formation. Importantly, none of these three compounds showed toxicity to a human cell line; therefore, EhFAEs are reasonable targets for developing new anti-amebiasis drugs and these compounds are promising leads for such drugs. Interestingly, in the Amoebazoan lineage, gain and loss of the genes encoding two different types of fatty acid elongase have occurred during evolution, which

http://prime.psc.riken.jp/menta.cgi/prime/drop_index, DM0056.

**Funding:** This work was supported by Grants-in-Aid for Scientific Research from the Ministry of Education, Culture, Sports, Science and Technology (MEXT) of Japan (18H04675 and 21K06992) to F.M., (21K18216, 24K02011, and 24H00043) to H.T., (19K07627) to H.Y., and (15H05897, 15H05898, 20H00495) to M.A., the AMED Japan Program for Infectious Diseases Research and Infrastructure (23wm0325036h0003) to F.M., H.T., H.Y. and M. A., JST ERATO "Arita Lipidome Atlas Project" (JPMJER2101 to H.T. and M.A.), AMED Moonshot Research and Development Program (JP22zf0127007 to M.A.), the National Cancer Center Research and Development Fund (2020-A-9, H.T.), AMED Brain/MINDS (JP15dm0207001, H. T.), the Kaketsuken Foundation to F.M., the Nagase Foundation to F.M., the Takeda Foundation to F.M., and Naitou Foundation to F.M. The funders had no role in study design, data collection and analysis, decision to publish, or preparation of the manuscript.

**Competing interests:** The authors have declared that no competing interests exist.

may be relevant to parasite adaptation. Acyl chain diversity in lipids is therefore a unique and indispensable feature for parasitic adaptation of *Entamoeba*.

## Author summary

*Entamoeba*, a protozoan parasite, thrives in its niche by alternating its form between a proliferative trophozoite and a dormant cyst. *Entamoeba* parasites were recently shown to generate lipid acyl chain diversity throughout their life cycle. Here, we provide insight into the mechanism generating this acyl chain diversity and its importance in maintaining the parasite's life cycle. We also show that the *E. histolytica* isozymes responsible for generating this diversity are reasonable targets for developing new anti-amebiasis drugs, and that the herbicides, flufenacet, cafenstrole, and fenoxasulfone, are promising leads for such drugs. Interestingly, gain and loss of the two genes encoding the isozymes controlling acyl chain diversity is plausibly related to *E. histolytica* host adaptation.

## Introduction

*Entamoeba histolytica* is a protozoan parasite belonging to the phylum *Amoebozoa*. This parasite causes amebiasis, which is a global public health problem [1,2]. The life cycle of *E. histolytica* consists of trophozoites and cysts [3,4]. Trophozoite proliferation is closely associated with the clinical manifestations of amebiasis whereas dormant cysts transmit the disease [5]. Metronidazole is the drug of choice for amebiasis treatment; however, adverse effects can occur and it may be insufficient to fully eradicate the infection [6]. Currently, no effective vaccines exist; therefore, new drugs are required.

Membrane properties of trophozoites and cysts are drastically different. The trophozoite is a proliferative amoeboid cell, in which vesicular trafficking and molecule transport across the plasma membrane are highly active. The cyst is a nonmotile round dormant cell, in which the number of intracellular vesicles is low and the plasma membrane is impermeable to substances [4,7].

We recently investigated changes in the membrane lipid profile throughout the life cycle of *Entamoeba* parasites (*E. histolytica* and *E. invadens*, which is used as an encystation model) using state-of-the-art liquid chromatography-mass spectrometry-based untargeted lipidomics [8]. This analysis revealed structural uniqueness of several lipid classes, including sphingolipids (SLs), glycerophospholipids (GPLs), and lysoglycerophospholipids (LGPLs). A hallmark of the structural uniqueness was that saturated or mono- or di-unsaturated very long (26–30 carbons) acyl chains and/or saturated medium (8–12 carbons) acyl chains were linked to these lipid classes. These very long acyl chains were detected in dihydroceramide (Cer-NDS), Cer phosphatidylethanolamine (CerPE), and Cer phosphatidylinositol (CerPI); phosphatidylcholine (PC), phosphatidylethanolamine (PE), phosphatidylserine (PS), and phosphatidylinositol (PI); and lysoPE, lysoPS, and lysoPI, while medium acyl chains were detected in PC, PE, PS, and PI [8,9]. Typically, acyl chains are incorporated into these lipids via acyl-coenzyme A (CoA), an activated fatty acid [10].

As a source of fatty acids, *Entamoeba* is thought to mainly rely on scavenging from the external milieu. This is because neither type I nor type II fatty acid synthases, which are responsible for *de novo* fatty acid synthesis, are genomically encoded [AmoebaDB release 68 (https://amoebadb.org/amoeba/app), and hereafter this database was used; [11]]. Very long

chain fatty acids (VLCFAs) and saturated medium-chain fatty acids, all of which can be converted to the above unique acyl chains, are, however, rare or at low levels in the host [Supplemental Data 1 in [12]]. In addition, fatty acid desaturase and enzymes involved in fatty acid oxidation, such as β-oxidation (two carbon units are released from substrate acyl-CoA every cycle), are not encoded in *Entamoeba* genomes [AmoebaDB, [11,13]]. *Entamoeba* may therefore use a fatty acid chain elongation system to provide required acyl-CoAs. Evidence suggests that this system does indeed function in *E. histolytica* [8].

Fatty acid chain elongation consists of four cyclic reactions (Fig 1A) [14,15]; (i) condensation of a substrate acyl-CoA with malonyl-CoA to produce β-ketoacyl-CoA; (ii) reduction of the β-ketoacyl-CoA to β-hydroxyacyl-CoA; (iii) dehydration of the β-hydroxyacyl-CoA to enoyl-CoA; (iv) reduction of the enoyl-CoA to a product acyl-CoA (two carbon units are added to the substrate acyl-CoA). The enzyme catalyzing the $1^{st}$ step in the cycle determines substrate acyl-CoA preference and is the rate-limiting enzyme. The genes encoding this enzyme are broadly conserved among eukaryotes [15]. Phylogenetic analysis indicates divergence of these enzymes into two groups. One group consists of proteins encoded in elongase (ELO) genes, which are usually present in non-photosynthetic protists, fungi, and mammals. The other group consists of proteins encoded by β-ketoacyl-CoA synthase/fatty acid elongase (KCS/FAE) genes, which are mostly present in photosynthetic protists and plants [16]. *E. histolytica*, a non-photosynthetic protist, is an exception that possesses five orthologous KCS/FAE genes, but no ELO gene [8,11,17]. Furthermore, the *E. histolytica* genome encodes three β-ketoacyl-CoA reductases, one β-hydroxyacyl-CoA dehydrase, and two *trans*-2-enoyl-CoA reductases, which are responsible for catalyzing reactions (ii) to (iv), respectively (Fig 1A) [AmoebaDB, [8,17]].

In this study, we characterized the first enzymes in the fatty acid elongation cycle in *E. histolytica* using a combined genetics and lipidomics approach. We also provide evidence that *E. histolytica* FAEs are reasonable targets for developing new anti-amebiasis drugs, and that flufenacet, cafenstrole, and fenoxasulfone are promising leads for such drugs. We also studied for the phylogeny of the first enzymes in eukaryotes, particularly in the phylum *Amoebozoa*.

## Results

### Fatty acid elongation capacity in *E. histolytica*

To show broad involvement of the fatty acid chain elongation system in *E. histolytica* lipid metabolism, trophozoites were metabolically labeled with [$^{13}$C18]-oleic acid ($C_{18:1}$), and then total lipids were analyzed by untargeted lipidomics. $^{13}$C-signals were detected in a series of mono-unsaturated fatty acids, FA 20:1, FA 22:1, FA 24:1, FA 26:1, FA 28:1, FA 30:1, and FA 32:1 (Fig 1B). $^{13}$C-signals were also incorporated into GPLs and SLs, including PE 18:1_30:1, PE 18:2_28:1, PE 10:0_30:1, PC 18:0_20:1, PS 9:0_28:1, PS 18:1_30:1, and Cer 18:0;O2/24:1 (http://prime.psc.riken.jp/menta.cgi/prime/drop_index, DM0056, Ameba 13C Agilent). These results indicate that oleic acid ($C_{18:1}$) is elongated and incorporated as a very long acyl chain into different lipid classes that are essential to maintain the life cycle of *Entamoeba* [8, 9]. Meanwhile, $^{13}$C-signals were not detected in FA 18:2 and FA 18:3 (Fig 1C), indicating that *Entamoeba* does not possess the ability to desaturate FA 18:1. This is consistent with the absence of desaturase genes in the *Entamoeba* genome [AmoebaDB, [11]].

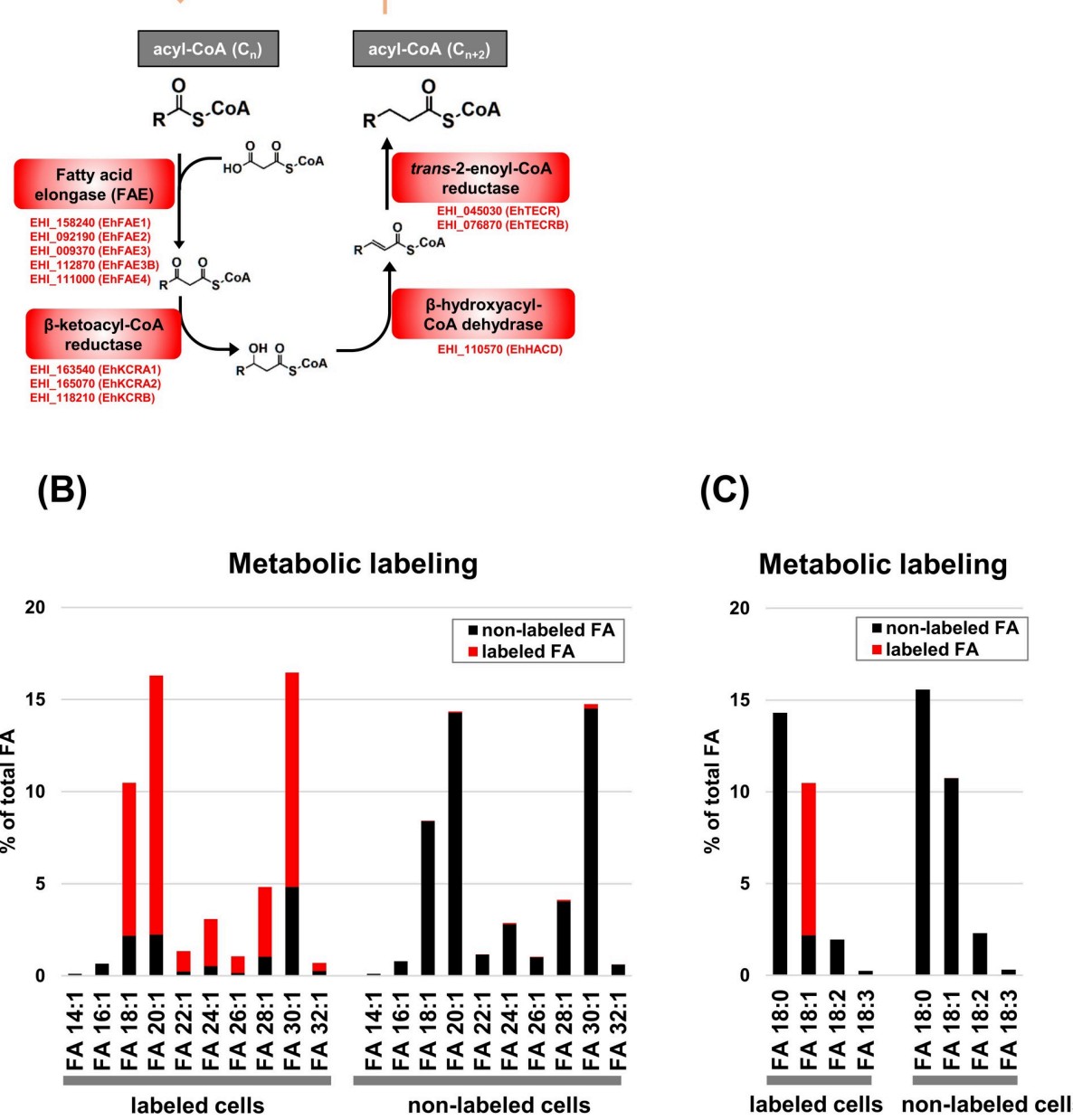

**Fig 1. *Entamoeba* capacity for fatty acid chain elongation.** (A) Biochemistry of the fatty acid elongation cycle. Deduced *E. histolytica* genes responsible for each step are indicated with AmoebaDB gene IDs. (B) Elongation of $^{13}$C-oleic acid to longer fatty acids. (C) Increase of unsaturation levels.

## Characterization of *Entamoeba* FAE isozymes based on the range of acyl chain lengths generated

The enzyme catalyzing the 1$^{st}$ step in the fatty acid elongation cycle (see Fig 1) is usually responsible for generating variations in the length of acyl chains that are linked to different

lipids [14,15]. *E. histolytica* and *E. invadens* possess five (*EhFAE1–4*, and *EhFAE3B*) and four (*EiFAE1–4*) orthologs of plant KCS/FAE genes, respectively [AmoebaDB, [8,11]]. These genes are named based on orthologous and paralogous relationships by phylogenetic analysis (Fig 2A). For instance, EhFAE3 and EiFAE3 are orthologous whereas EhFAE3 and EhFAE3B are

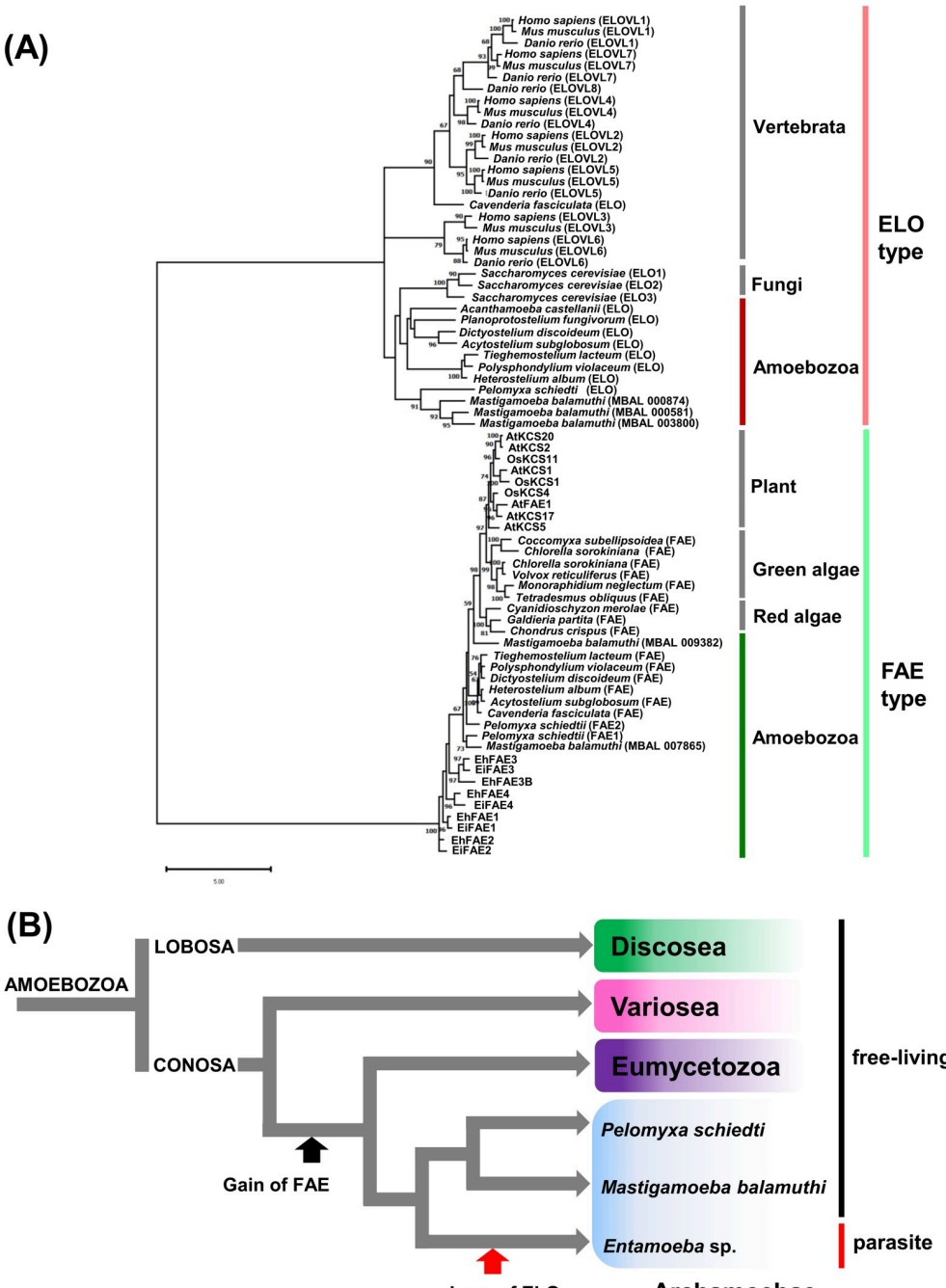

**Fig 2. Evolution of the 1st enzyme in the fatty acid elongation cycle in eukaryotes.** (A) Phylogenetic tree of the 1st enzyme in the fatty acid elongation cycle. Bootstrap proportion (BP) values are attached to the internal branches. Branches with less than 50% BP support are unmarked. (B) A plausible scenario for gain of FAE gene and loss of ELO gene in the Amoebozoan lineage evolution.

paralogous. Hereafter, the products of plant KCS/FAE orthologs are termed FAEs whereas those of mammalian ELO orthologs are termed ELOs. To identify the range of acyl chain length for which each *Entamoeba* FAE is responsible, we exploited an approach combining genetics and lipidomics. The genetic approach included gene overexpression and knockdown using *E. histolytica* as a host because this enables substrate preference and product spectra of EhFAE isozymes to be analyzed in natural substrate conditions.

Each EhFAE gene was separately overexpressed in *E. histolytica* transformants, namely, EhFAE1-HA to EhFAE4-HA, and EhFAE3B-HA, to produce hemagglutinin (HA)-tagged proteins. In all strains, only the targeted EhFAE gene was selectively upregulated (S1A Fig) and its product was detected by an anti-HA antibody (S2 Fig). The levels of EhFAE1-HA and EhFAE3B-HA were higher than those of the other three EhFAEs and the level of EhFAE3-HA was lower than those of the other four EhFAEs. (S2 Fig). Total lipids were extracted from each transformant and analyzed by untargeted lipidomics (Fig 3A and 3B). In EhFAE1-HA, the levels of saturated and unsaturated fatty acids (carbon numbers greater than 30) were increased (FAE1 in Fig 3A) and those of Cers containing saturated (28 carbons) and mono- and di-unsaturated (30 carbons) acyl chains were increased (FAE1 in S3 Fig). In EhFAE2-HA, levels of saturated and mono- and tri-unsaturated fatty acids (carbon numbers 20–28) were increased (FAE2 in Fig 3A). In EhFAE3-HA, levels of 9–11 carbon saturated fatty acids were increased (FAE3 in Fig 3A) and those of Cers containing 30 carbon mono-unsaturated acyl chains were increased (FAE3 in S3 Fig). In EhFAE3B-HA, levels of FA 10:0 and FA 11:0 were increased (FAE3B in Fig 3A) and those of Cers containing 28 carbon saturated and 30 carbons mono-unsaturated acyl chains were increased (FAE3B in S3 Fig). In EhFAE4-HA, levels of 20–26 carbon mono-unsaturated fatty acids were increased (FAE4 in Fig 3A). Similar patterns as those above were observed for PEs, with increased levels of acyl chains of different lengths in each of EhFAE1-HA to EhFAE3-HA, and EhFAE3B-HA (Fig 3B). These results indicate that variations in acyl chains were generated by ectopic overexpression of EhFAE isozymes. EhFAE1 is involved in the production of >C28-acyl chains. EhFAE2 is involved in the production of C10-, and C20–30-acyl chains. EhFAE3 is involved in the production of C9-, C10-, and C30-di-unsaturated-acyl chains. EhFAE3B is involved in the production of C10-, C11-, and C30-mono-unsaturated-acyl chains. EhFAE4 is involved in the production of C22–26-mono-unsaturated-acyl chains.

A gene knockdown experiment was also performed; each EhFAE gene was knocked down in *E. histolytica* G3 strain to produce "EhFAE1–4gs and EhFAE3Bgs". EhFAE2, EhFAE3, and EhFAE3Bgs could be obtained, in which only the targeted gene among the five EhFAEs (*EhFAE1–4*, and *EhFAE3B*) was knocked down (S1B Fig); however, EhFAE1 and EhFAE4gs could not be obtained, probably because these knockdowns caused lethality. The lipidomic profiles in all three EhFAEgs and mock transformants were then individually determined (S4 Fig). Lipidomics data of gene knockdown strains provided additional evidence for the steps catalyzed by EhFAEs. In gsEhFAE2, the levels of FA 9:0, 12:0, and 17:1 and of PSs containing unsaturated acyl chains of more than 30 carbon were decreased (gsFAE1 in S4A and S4B Fig). In gsEhFAE3B, levels of FA 9:0 and FA 12:0 and of PSs and PEs containing 10 carbon acyl chains were decreased (gsFAE3B in S4A–S4C Fig). The above genetic and lipidomic results collectively enable the acyl chain elongation steps mediated by *Entamoeba* FAEs to be deduced (Fig 4).

## Plant FAE inhibitors impair *Entamoeba* cell growth and encystation

*Entamoeba* parasites are Archamoebea in the phylum *Amoebozoa* [18,19] and atypically possess orthologs of plant KCS/FAE genes [11]. This unusual feature is also found in other

**(A) FFA**

**(B) PE**

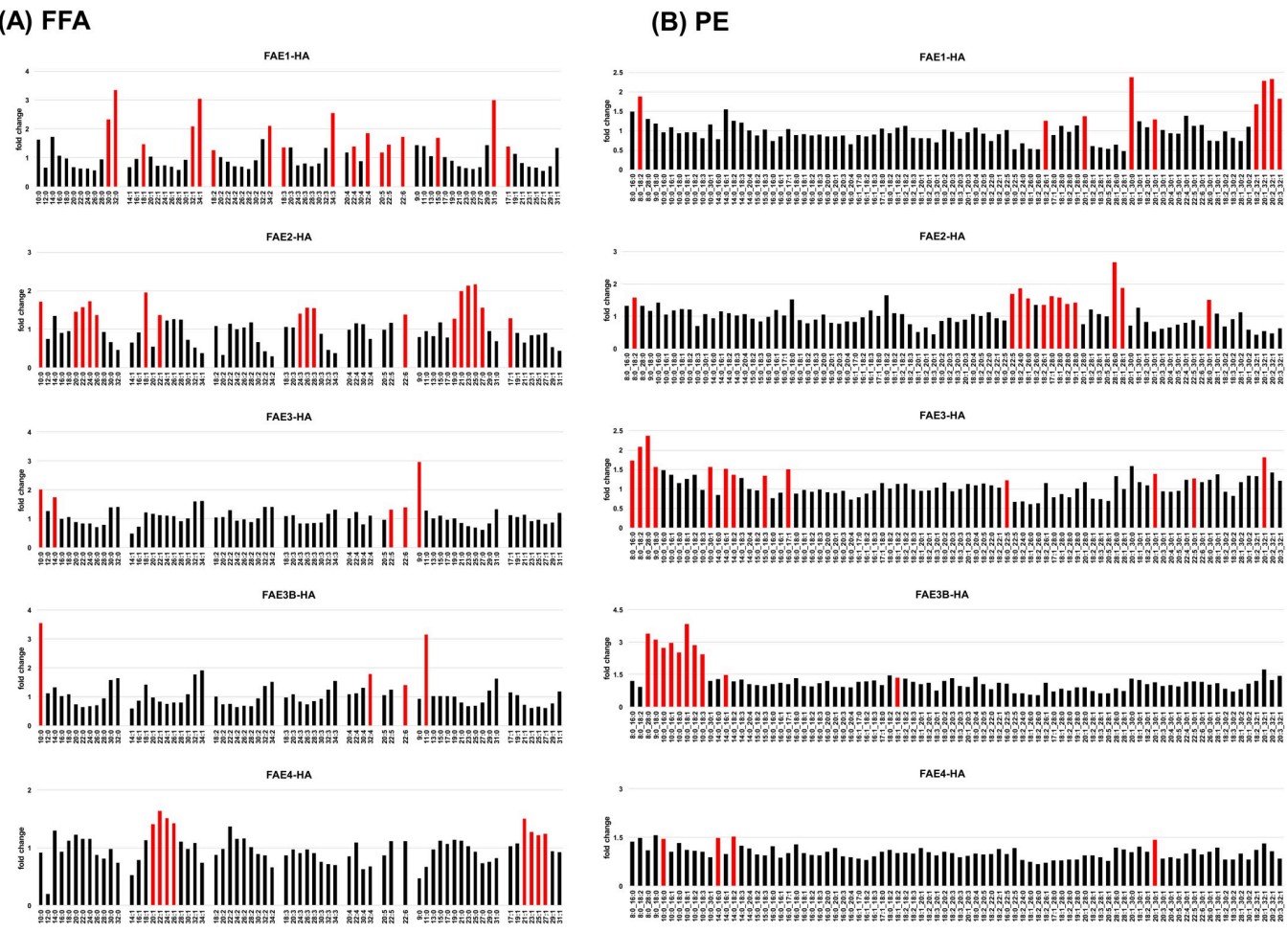

**Fig 3. Changes in the profiles of fatty acids (A) or acyl chains in PE (B) by overexpressing each EhFAE gene in *E. histolytica*.** Signal intensity levels are shown as fold change to that of the control strain. Lipid species that showed a statistically significant (P<0.05) and >1.2-fold increase are indicated by red bars.

Amoebozoan organisms, such as *Dictyostelium discoideum* and *Mastigamoeba balamuthi*, which belong to Eumycetozoa and Archamoebea subclades, respectively. Of note, these organisms also possess mammalian ELO gene orthologs. Meanwhile, *Planoprotostelium fungivorum*, which belongs to the Variosea class [18, 19], only possesses the mammalian ELO gene ortholog, which is typically seen in non-photosynthetic protists (Fig 2A). The fact that *Entamoeba* only possesses plant-type FAE genes (S5 Fig) together with the phylogeny of Amoebozoa [Fig 1 in [18]] indicate that the common ancestor of Eumycetozoa and Archamoebae acquired FAE genes from plants or red/green alga, possibly by lateral gene transfer, after branching of the Variosea lineage. Among the Archamoebea, the Entamoeba lineage has then undergone secondary loss of mammalian-type ELO genes during the course of evolution (Fig 2B). Importantly, *Entamoeba* parasites rely on FAEs only and not on ELOs for fatty acid chain elongation [14,15,20]. Inhibiting FAEs may therefore halt the parasite life cycle.

To prove our assumption, flufenacet, cafenstrole, and fenoxasulfone, classified by the Herbicide Resistance Action Committee (https://www.hracglobal.com/) as K3 group herbicides and which are commercially available (FUJIFILM Wako; Osaka, Japan), were used because K3 group compounds inhibit VLCFA biosynthesis in plant cells [21,22]. Furthermore, flufenacet and cafenstrole broadly inhibit FAE isozymes in *Arabidopsis thaliana* [21]. Effectiveness of

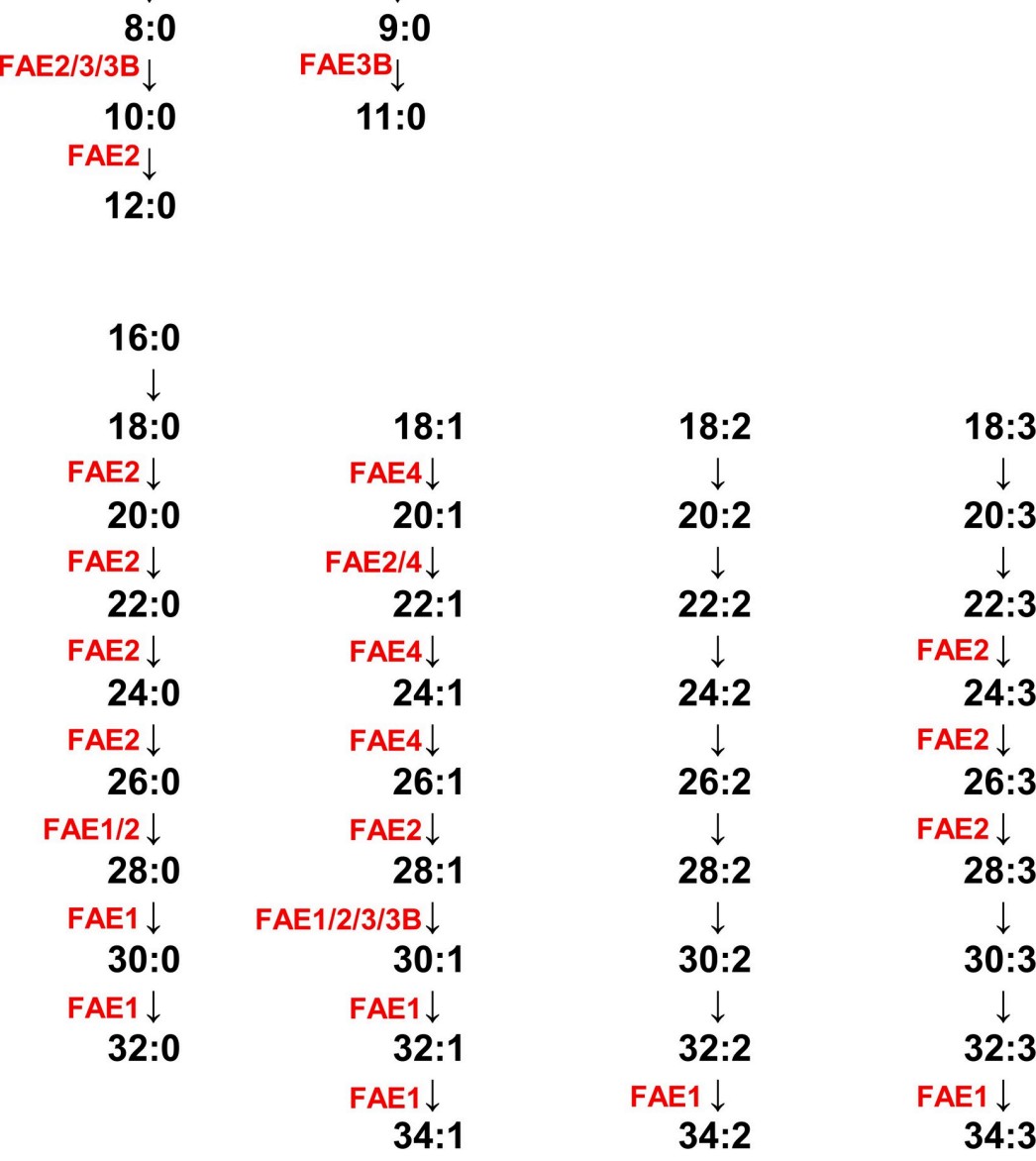

**Fig 4. Deduced fatty acid elongation pathway in *E. histolytica*.**

these compounds against trophozoites and cysts was separately assessed by flow cytometry as described previously [23]; propidium iodide, a membrane impermeable dye that binds to DNA, was used to accurately count intact cells by eliminating dead trophozoites (propidium iodide+). Calcofluor (CF), a dye that specifically binds to chitin wall, and Evans blue (EB), a dye that binds to intact cell membrane, were both used to detect mature cysts (CF+/EB-). Flufenacet, cafenstrole, and fenoxasulfone all dose-dependently inhibited trophozoite proliferation of *E. histolytica* with $IC_{50}$s of $1.6 \pm 0.7$ μM, $70 \pm 26$ μM, and $7.0 \pm 0.5$ μM, respectively (Fig 5A and 5B). These three compounds also inhibited cyst formation in *in vitro E. invadens* cultures with $IC_{50}$s of $0.48 \pm 0.11$ μM, $0.10 \pm 0.02$ μM, and $1.7 \pm 0.3$ μM, respectively (Fig 5A and 5C).

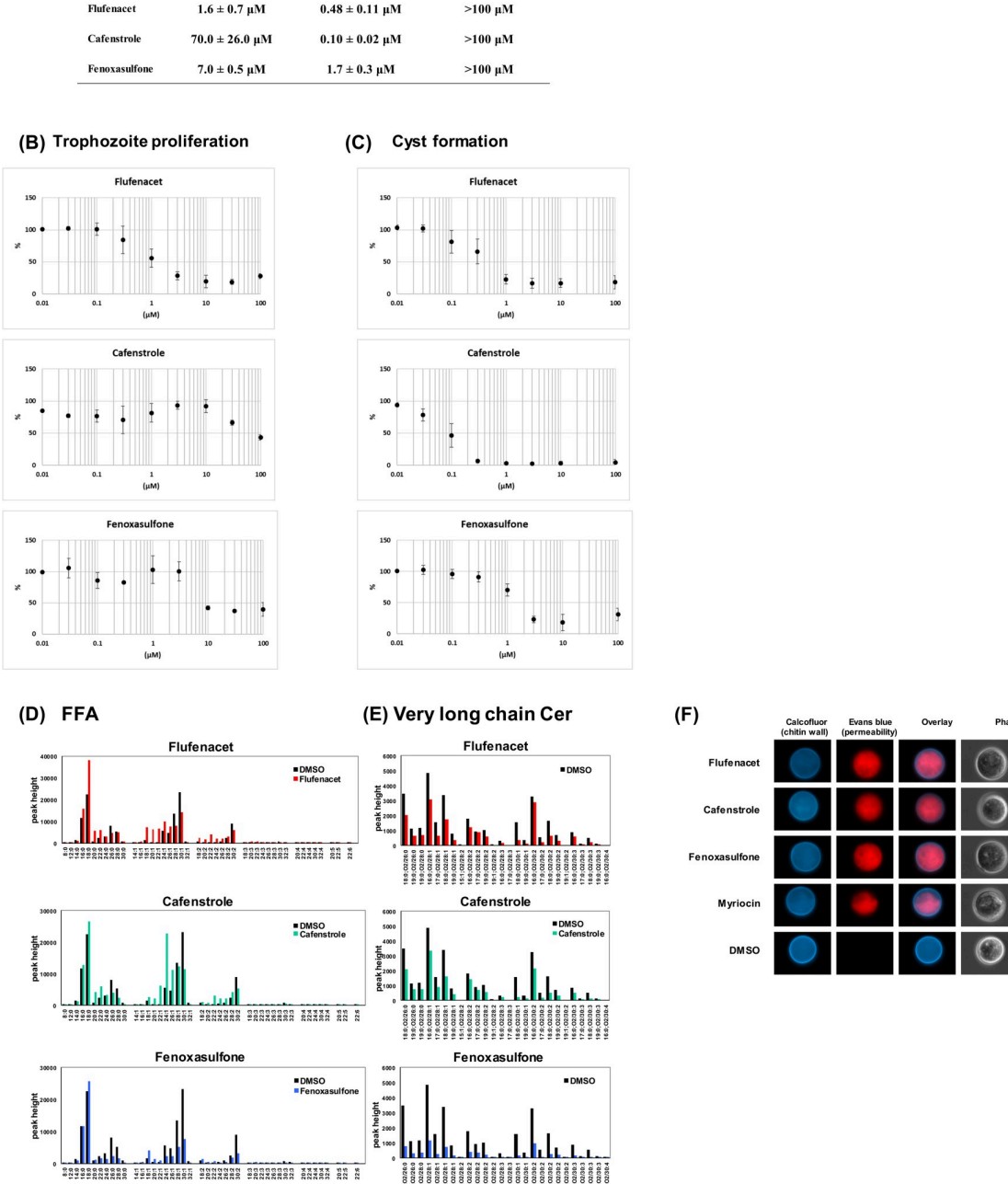

**Fig 5. Effects of flufenacet, cafenstrole, and fenoxasulfone on crucial processes in the *Entamoeba* life cycle and on human cells.** (A) Summary of flufenacet, cafenstrole, and fenoxasulfone $IC_{50}$ values for trophozoite proliferation and cyst formation, and of their cytotoxicity to human cells. $IC_{50}$ values for proliferating *E. histolytica* trophozoites and encysting *E. invadens* cells, and cytotoxicity to human foreskin fibroblasts (HFFs) were determined by three independent experiments and are expressed as the average ± SD. (B, C) Inhibition curves of each compound against trophozoites and encysting cells. Data are indicated by filled circles (averages) with error bars (SDs). (D, E) Untargeted lipidomics. The profiles of fatty acids (D) and very long chain Cer-NDSs (E). Representative data are shown from four independent experiments. (F) Fluorescence microscopy. Myriocin, which halts *de novo* ceramide synthesis by inhibiting the 1st enzyme in ceramide synthesis, serine palmitoyltransferase, was used as a control.

Notably, at 100 μM, none of the three compounds showed cytotoxicity against a human fore-skin fibroblast (HFF) cell line (Fig 5A).

We then attempted to identify the targets and effects of flufenacet, cafenstrole, and fenoxa-sulfone in *Entamoeba* encysting cells by untargeted lipidomics and fluorescence microscopy. Encysting *E. invadens* cells were treated with each of flufenacet, cafenstrole, and fenoxasulfone at 1, 0.3, and 3 μM (see Fig 5C), respectively, for 48 h. The treated cells were then collected for untargeted lipidomics. Each compound treatment significantly reduced cyst formation rates; relative values of flufenacet, cafenstrole, and fenoxasulfone treatment to that of dimethyl sulf-oxide (DMSO) control were ~25, ~7, and ~20%, respectively. In the treated cells, levels of VLCFAs, such as FA 26:0, FA 28:0, FA 30:0, FA 28:1, FA 30:1, FA 32:1, and FA 30:2, were sig-nificantly reduced. Conversely, levels of long-chain fatty acids (FA 20:0, FA 20:1, FA 22:1, FA 20:2, and FA 22:2) were substantially increased (Fig 5D). Similarly, levels of very long acyl chains (>26 carbons) linked to dihydroceramides, such as Cer 18:0;O2/30:1, Cer 19:0;O2/30:1, and Cer 18:0;O2/30:2, were reduced (Fig 5E). Overall, the abundance of lipid species was sig-nificantly and similarly reduced in all treated cells.

In encysting *E. invadens* cells treated with flufenacet, cafenstrole, and fenoxasulfone at 100, 3, and 100 μM, respectively (see Fig 5C), chitins were synthesized and placed in the cyst wall at similar levels to that in DMSO-treated control cells (Fig 5F), indicating that *E. invadens* FAEs are not involved in synthesis of the cyst wall, a prerequisite for *Entamoeba* parasites becoming dormant [4]. However, EB fluorescence, an indicator of membrane permeability, abnormally accumulated in the treated cells (Fig 5F), indicating that FAE inhibition increased the mem-brane permeability of the encysting cells. For *Entamoeba* parasites to become dormant it is essential to generate membrane impermeability. Membrane impermeability of dormant cysts is established by significant elevation and accumulation of very long chain Cer-NDSs (dihy-droceramides) during encystation, which is regulated by cholesteryl sulfate [8, 24]. We there-fore suggest that destruction of membrane impermeability in *Entamoeba* parasites by flufenacet, cafenstrole, and fenoxasulfone results from impaired synthesis of very long chain Cer-NDSs via inhibition of FAE isozymes. Interestingly, there is a physiological resemblance in the membrane impermeability of mature *Entamoeba* cysts and the skin barrier function of retaining water and lipids in humans and mice. To exert these barrier functions, lipid lamellae, which are highly abundant in very long chain Cer-NSs (ceramides) (FA portion of C24–C30) and acyl-Cer-NSs (FA portion of C30–C36), have crucial roles [20].

Taken together, these results indicate that flufenacet, cafenstrole, and fenoxasulfone all tar-get *Entamoeba* FAE isozymes, and that acyl chain variations in different lipid classes generated via the FAE isozymes are essential for the *Entamoeba* life cycle.

## Discussion

Acyl chain diversity in GPLs and SLs is important for various membrane functions. In *Ent-amoeba* parasites (*E. histolytica* and *E. invadens*, an encystation model), unique acyl chains were recently shown by untargeted lipidomics to be linked to several GPL, SL, and LGPL lipid subclasses. These were saturated or mono- or di-unsaturated very long (26–30 carbons) acyl chains and/or saturated medium acyl chains [8,9], which are rare in mammals [Supplemental Data 1 in [12]]. Here, to elucidate the molecular mechanism underlying the generation of acyl chain diversity in *Entamoeba* membranes, we characterized *Entamoeba* FAE isozymes, the 1st enzyme in the fatty acid elongation cycle.

The capacity for fatty acid elongation in *Entamoeba* parasites was initially confirmed by [13]C-oleic acid ($C_{18:1}$) metabolic labeling of *E. histolytica* trophozoites; [13]C-mono-unsaturated VLCFAs (more than 20 carbons) were detected while any of [13]C-saturated and di-, tri-, or

polyunsaturated fatty acids were not. Degraded products of $^{13}$C-$C_{18:1}$, such as $^{13}$C-$C_{14:1}$ and -$C_{16:1}$, were not detected either. These results are consistent with the absence of genes encoding fatty acid desaturase and enzymes for β-oxidation in *Entamoeba* genomes. Furthermore, *Entamoeba* relies solely on scavenging from the external milieu for required fatty acids because *de novo* fatty acid synthesis systems are not encoded in *Entamoeba* genomes [11]. These findings lead us to predict that in *Entamoeba*, the series of VLCFAs are synthesized from the major fatty acids available in the host via the fatty acid elongation cycle [Supplemental Data 1 in [12]]. For instance, in *E. histolytica*, palmitic ($C_{16:0}$) and stearic ($C_{18:0}$) acids are elongated to $C_{26:0}$, $C_{28:0}$, and $C_{30:0}$. Oleic acid ($C_{18:1}$) is elongated to $C_{20:1}$, $C_{22:1}$, $C_{24:1}$, $C_{26:1}$, $C_{28:1}$, and $C_{30:1}$. Linoleic acid ($C_{18:2}$) is elongated to $C_{24:2}$, $C_{28:2}$, and $C_{30:2}$. Importantly, metabolic labeling also identified $^{13}$C-labeled GPLs, which carry mono-unsaturated very long acyl chains, indicating that VLCFAs produced by the elongation cycle are incorporated into acyl chains. It is therefore very likely that the *E. histolytica* fatty acid elongation cycle is critical in generating the diversity of very long acyl chains among membrane lipids using palmitic ($C_{16:0}$), stearic ($C_{18:0}$), oleic ($C_{18:1}$), and linoleic ($C_{18:2}$) acids as initial substrates. This is consistent with our previous untargeted lipidomics data which showed abundant SLs and GPLs containing saturated-, and mono- and di-unsaturated very long (26–30 carbons) acyl chains, but few polyunsaturated very long acyl chains [8,9].

The present approach of combining genetics and lipidomics enables the product spectra and substrate preferences of *Entamoeba* FAE isozymes to be detected (Fig 4). *Entamoeba* parasites use FAE2 and FAE4 and host-derived fatty acids, mainly $C_{16:0}$, $C_{18:0}$, $C_{18:1}$, and $C_{18:2}$, as substrates for the production of C20–C30-CoAs. These products are further elongated to very long acyl chains (more than 28 carbons) via FAE1 and FAE2. However, not all the potential steps involved in elongating the major four host-derived fatty acids, for which the FAE isozymes are responsible, could be deduced. This is because of integrated reaction chains and links to other lipid metabolic pathways. That is, a product from a certain cycle becomes a substrate for a subsequent cycle and/or is introduced as an acyl chain into various lipids of other pathways. Nevertheless, similar trends of increased and decreased acyl chain lengths can be detected in lipids, although the change levels of some acyl chains were small.

Interestingly, saturated medium-chain fatty acids (8–12 carbons) were also produced by FAE2 and FAE3 (and also FAE3B in *E. histolytica*) in the fatty acid elongation cycle. This product spectrum confirms our previous lipidomics data that showed saturated medium acyl chains linked to PE, PS, and PI [8,9]. The degradation of host-derived fatty acids, such as $C_{14:0}$, $C_{16:0}$, and $C_{18:0}$, to provide fatty acids with 8–12 carbons, is very unlikely because *Entamoeba* loses the capacity for β-oxidation [11]. This assumption raises an important question; what is the initial substrate for the production of saturated medium-chain acyl-CoA? The most reasonable answer is that *Entamoeba* FAE2 and FAE3 (and FAE3B) uses acetyl-CoA (two carbon acyl-CoA) and/or butyryl-CoA (four carbon acyl-CoA) as the initial substrate. Furthermore, *Entamoeba* FAE2 and FAE3 (and FAE3B) may also be involved in the production of saturated odd-numbered medium acyl chains, which were detected in our previous untargeted lipidomics analysis [8,9]. In this case, propyl-CoA (three carbon acyl-CoA) could be the initial substrate.

As described above (see Fig 2A and 2B), in a monophyletic tree of Amoebozoa [18,19] a common ancestor of two major subclades, Eumycetozoa and Archamoebae, is assumed to have acquired an FAE gene from plants or red/green alga after branching of the Variosea lineage. In addition, *Entamoeba* FAE2 and FAE3 (and also FAE3B in *E. histolytica*) are involved in the generation of saturated medium acyl chains, whereas *A. thaliana* possesses FAE isozymes involved in the synthesis of VLCFAs, similar to *Entamoeba* FAE1, 2, and 4 [21]. These findings raise an interesting and important question as to whether *Dictyostelium*, *Mastigamoeba*, and

plants and red/green alga possess similar FAE(s) to *Entamoeba* FAE2 and FAE3 (and FAE3B). If they do not, having the ability to synthesize saturated medium acyl chains may be beneficial for *Entamoeba* parasites to adapt to their hosts.

Among Amoebozoan species, only *Entamoeba*, a parasitic species, lost ELO genes and relies solely on FAEs. This may be related to the fact that the source of fatty acids is significantly different between parasitic and free-living organisms. Therefore, secondary loss of ELO genes is plausibly responsible for parasitic adaptation. For a deep understanding of FAE and ELO gene evolution, it is necessary to characterize FAEs, ELOs, and lipidomes in other Amoebozoan organisms and also to investigate their sources of fatty acids.

In this study, flufenacet, cafenstrole, and fenoxasulfone, inhibitors of plant FAE isozymes, impaired both trophozoite proliferation of *E. histolytica* and cyst formation of *E. invadens*. Comparing the potency ratios of the chemicals ($IC_{50, \text{ trophozoite proliferation}}/IC_{50, \text{ cyst formation}}$) (see Fig 5A), cafenstrole showed a much higher value (~700) than flufenacet (~3) and fenoxasulfone (~6). This may attribute to difference in the permeability of cafenstrole to cells at different stages and/or from different species. *Entamoeba* cells treated with any of these three compounds consistently displayed different lipid profiles from that of DMSO-treated control cells. Abundances of lipid species containing very long acyl chains were dramatically decreased in FAE inhibitor-treated cells. These results together with the deduced fatty acid elongation pathway in *Entamoeba* (see Fig 4) indicate that all three compounds target *Entamoeba* FAE1, 2, and 4, and that *Entamoeba* FAE1, 2, and 4 isozymes are essential for the *Entamoeba* life cycle. Lipids have been shown to be key molecules in *E. histolytica* virulence. For instance, lysoPC, which is formed from PC in *E. histolytica*, is a potent lytic molecule against host target cells. Furthermore, for host invasion, *E. histolytica* has a self-protection system that generates diversity in the composition of the membrane lipid to confer resistance to glycosidases and cysteine proteases, which are secreted by the parasites and important in the destruction of host cells/tissues [17]. Acyl chain diversity may be crucial for these functions; therefore, it is plausible that the above three herbicides also affect *E. histolytica* virulence by inhibiting EhFAE isozymes.

None of the three compounds significantly impaired the proliferation of human HFFs, indicating high selectivity toward *Entamoeba* parasites. This result indicates that there are no targets of these compounds in HFFs. In contrast to *Entamoeba* parasites, human cells use ELO isozymes, not FAEs, as the 1st enzyme in the fatty acid elongation cycle, and generate very long acyl chains via the isozymes, elongases of very long chain fatty acids (ELOVLs) [14,15]. Importantly, several ELOVLs are functional equivalents to *Saccharomyces cerevisiae* ELO isozymes, and some *S. cerevisiae* counterparts of human ELOVLs have high resistance to K3 group herbicides [21]. Therefore, this study gives a mechanistic explanation for the potency of these compounds toward *Entamoeba* parasites and for their very low cytotoxicity to human cells. This study also shows that *E. histolytica* FAE1 and -2 (and also most likely -4) are reasonable targets for the development of new drugs against amebiasis, and that flufenacet, cafenstrole, and fenoxasulfone are promising leads for such drugs.

In conclusion, this study provides mechanistic insight into the fatty acid elongation cycle that generates unique acyl chain diversity in *Entamoeba* lipids and indicates reasonable targets and promising leads for the development of new anti-amebiasis drugs. This study also discusses the relevance of FAE gene gain and ELO gene loss in the evolution of parasitic adaptation. Our findings contribute to a deeper understanding of *Entamoeba* lipid biochemistry, physiology, and evolution and also provide a basis for pharmaceutical intervention to treat the global public health problem of amebiasis.

## Materials and methods

### Parasite cultures, induction of encystation, flow cytometry assays for trophozoite proliferation and cyst formation, and fluorescence microscopy

*E. histolytica* (HM-1:IMSS cl6) and (G3) were routinely maintained in Diamond BI-S-33 medium at 37˚C as described previously [25]. Maintenance of *E. invadens* (IP-1) in routine cultures and induction of encystation were performed essentially as described previously [26]. For induction of encystation, 'stationary phase' trophozoites were harvested and transferred to encystation medium at a final density of $6 \times 10^5$/mL.

Assays for *E. histolytica* trophozoite proliferation and *E. invadens* cyst formation were performed essentially as described previously with modifications; in the trophozoite assay, the initial cell number and culture volume in each well were $2 \times 10^4$ and 240 μL, respectively. A BDCelesta flow cytometer (Becton Dickinson, NJ, USA) was used in place of a MACSQuant (Miltenyi Biotec; Gladbach, Germany). Calcofluor (CF) was excited using a 405 nm laser, and the fluorescence emission was collected using a 450/40 filter. EB and propidium iodide were both excited using a 488 nm laser and the fluorescence emission was collected using a 610/20 nm filter. The processing volumes were set at 20 μL and 30 μL for trophozoite proliferation assays and cyst formation assays, respectively. Samples were automatically injected from the 120 μL of fluorescent dye(s)-treated cell suspension in each well. Obtained data were analyzed using FlowJo software (Becton Dickinson, NJ, USA) as described previously [23]. For fluorescence microscopy, portions of the flow cytometry samples (compound treated and control) were examined as described previously [8].

### Metabolic labeling of *E. histolytica* and lipid analysis

*E. histolytica* trophozoites ($3 \times 10^5$ cells) were suspended in 6 mL medium supplemented with [$^{13}$C18]-oleic acid ($C_{18:1}$) (TAIYO NIPPON SANSO, Japan) at 17.5 μg/ml and the culture was incubated at 37˚C for 24 hr. Cells were pelleted by centrifugation at $750 \times g$ for 5 min at 4˚C, washed with 6 ml phosphate buffered saline (PBS), and resuspended in 2 mL PBS. The cell suspension was divided into two 1.5-ml sample tubes, and then cells were repelleted by centrifugation. These cell pellets were kept at −80˚C until use. The lipid extraction and LC-MS/MS protocols were described in S1 Text.

### Overexpression or knockdown of each EhFAE gene in *E. histolytica*

PCR amplicons encoding full-length or N-terminal (1-430bp) of target EhFAEs were obtained using suitable primer sets (S1 Table), digested with *Nhe*I and *Bgl*II (or *Bam*HI) or *Stu*I and *Sac*I and inserted into the corresponding sites of pEhEx-m-HA or pSAP2-g-multi, respectively [8,27]. After sequencing, a correct plasmid from each construction was introduced into *E. histolytica* trophozoites (HM-1:IMSS cl6 for gene overexpression or G3 for gene knockdown) and stable transformants were established. The established transformants were all maintained under G418 selection at 20 μg/ml as described previously [27]. The levels of target gene overexpression or knockdown in the established transformants were evaluated by real-time qRT-PCR using suitable primer sets (S1 Table) as previously described [8].

### Western blotting

Whole cell lysates of each EhFAE gene overexpressing strain were analyzed by western blotting essentially as previously described [28]. Loaded amount was adjusted by cell number ($3 \times 10^4$ cells per lane). A mouse monoclonal anti-HA antibody, HA-7 (Sigma-Aldrich, St. Louis, MO, USA), was used at 10,000-fold and 2,000-fold dilutions in S2A and S2B Fig, respectively. A

secondary antibody, HRP-conjugated sheep anti-mouse IgG (Cytiva, Tokyo, Japan), was used at 10,000-fold and 3,000-fold dilutions in S2A and S2B Fig, respectively. ImmunoStar Zeta and Immunostar LD (Fujifirm Wako, Osaka, Japan) was used for the detection in S2A and S2B Fig, respectively.

## LC-MS/MS-based lipidomics

*E. histolytica* transformants were analyzed as described previously [8]. *E. invadens* cyst formation was induced as described previously except that flufenacet, cafenstrole, and fenoxasulfone at 1, 0.3, and 3 μM (~$IC_{50}$s), respectively, and DMSO (solvent control) at 1% were added individually to cultures. Briefly, *E. invadens* trophozoites suspended in encystation medium ($6 \times 10^5$ cells/mL) were seeded in 96-well culture plates (0.24 mL per well) and sealed as described previously [23]. Then plates were incubated at 26˚C for 48 hr. Cells from seven wells were collected in a single 1.5-mL tube using 1 mL PBS, centrifuged at $800 \times g$ for 5 min at 4˚C, and washed with 1 ml PBS. Cells were resuspended in 1 mL PBS, divided into two 1.5-ml tubes, and repelleted by centrifugation. Cell pellets were kept at −80˚C until use. The lipid extraction and LC-MS/MS protocols were described in S1 Text.

## Statistics

Student's t-test was used to analyze the data in Figs 3, S3, S1A, and S1B. In Figs 3 and S3, differences between the signal intensity levels of lipid species in each transformant and those in the control strain were individually analyzed. Statistically significant increases in the level of each lipid species in each transformant were determined relative to levels in control cells. Lipidomics data from three independent experiments were used.

In S1A and S1B Fig, the increased (A) or decreased (B) mRNA level of the targeted FAE gene relative to the mRNA levels of the other FAE genes was individually analyzed in each transformant. Data from duplicates of three independent experiments were used.

## Cytotoxicity assay

An HFF cell line was purchased from ATCC (Washington, DC, NW, USA) and routinely maintained. A WST-1-based cytotoxicity assay was performed as previously described [29].

## Evolutionary analysis

All FAEs listed in Amoebozoa were obtained by Blast search (https://blast.ncbi.nlm.nih.gov/Blast.cgi?PAGE=Proteins) using *Entamoeba* FAE1 as query, whereas a representative ELO was used from publicly available databases. All FAEs or ELOs of *M. balamuthi* were obtained by Blast search (AmoebaDB) using *Entamoeba* FAE1 or *Saccharomyces* ELO1 as query. The other FAEs and ELOs were selected from publicly available databases, which are registered as FAEs and ELOs (see S2 Table). The evolutionary history was inferred by using the Maximum Likelihood method and Le_Gascuel_2008 model [30]. The tree with the highest log likelihood (-47653.57) is shown. The percentage of trees in which the associated taxa clustered together is shown next to the branches. Initial tree(s) for the heuristic search were obtained automatically by applying Neighbor-Join and BioNJ algorithms to a matrix of pairwise distances estimated using the JTT model, and then selecting the topology with superior log likelihood value. A discrete Gamma distribution was used to model evolutionary rate differences among sites [5 categories (+G, parameter = 2.1169)]. The tree is drawn to scale, with branch lengths measured in the number of substitutions per site. This analysis involved 73 amino acid sequences (Tabel

S2). There were a total of 1322 positions in the final dataset. Evolutionary analyses were conducted in MEGA11 [31].

## Supporting information

**S1 Text. LC-MS/MS-based lipidomics.**
(DOCX)

**S1 Fig. The level of gene overexpression or knockdown in each transformant.** (A, B) The transcription levels of FAE genes in each transformant were quantified by quantitative reverse transcription-PCR (qRT-PCR) using the EhRNApol gene as a control. The average (bars) and standard deviation (SD) from the average (error bars) were calculated from the data obtained from duplicates of three independent experiments. *, P<0.05; **, P<0.01.
(TIF)

**S2 Fig. Western blotting of each overexpressed gene product in transformant lysate using a rabbit anti-HA antibody.** (A) All transformants and control lysates were loaded. (B) FAE3-HA and control cell lysates were only loaded to enhance the signal.
(TIF)

**S3 Fig. Changes in the profiles of acyl chains in Cer.** Continued from Fig 3.
(TIF)

**S4 Fig. Changes in the profiles of fatty acids (A) or acyl chains in PS (B) and PE (C) by knocking down each EhFAE gene in *E. histolytica*.** Signal intensity levels are shown as fold change to that of the control strain. Lipid species that showed a statistically significant (P<0.05) and >20% decrease are indicated by red bars.
(TIF)

**S5 Fig. Fatty acid elongase types present in organisms in the phylum *Amoebozoa*.**
(TIF)

**S1 Table. List of primers used for plasmid constructions and qRT-PCR.**
(TIF)

**S2 Table. List of ELOs and FAEs from prokaryotes to eukaryotes.** GenBank accession numbers for ELOs and FAEs, which were used to infer the phylogenetic relationship shown in Fig 2A, are listed.
(TIF)

## Acknowledgments

We thank Ms. Aya Hori, Kyoko Nagatomo, Ayumi Fujimatsu, Akemi Ura, Ritsuko Yoshida, Minako Suzuki, Mami Ohtsubo, and Mayuko Fujisawa for technical assistance. We thank Jeremy Allen, PhD, from Edanz (https://jp.edanz.com/ac) for editing a draft of this manuscript.

## Author Contributions

**Conceptualization:** Fumika Mi-ichi, Hiroshi Tsugawa, Makoto Arita.

**Data curation:** Fumika Mi-ichi, Hiroshi Tsugawa.

**Formal analysis:** Fumika Mi-ichi, Hiroshi Tsugawa, Tam Kha Vo, Yuto Kurizaki, Makoto Arita.

**Funding acquisition:** Fumika Mi-ichi, Hiroshi Tsugawa, Hiroki Yoshida, Makoto Arita.

**Investigation:** Fumika Mi-ichi, Hiroshi Tsugawa, Tam Kha Vo, Yuto Kurizaki, Hiroki Yoshida, Makoto Arita.

**Methodology:** Hiroshi Tsugawa, Yuto Kurizaki.

**Project administration:** Fumika Mi-ichi.

**Software:** Hiroshi Tsugawa.

**Supervision:** Fumika Mi-ichi.

**Validation:** Fumika Mi-ichi, Hiroshi Tsugawa, Tam Kha Vo, Yuto Kurizaki, Hiroki Yoshida, Makoto Arita.

**Visualization:** Fumika Mi-ichi, Hiroshi Tsugawa, Tam Kha Vo.

**Writing – original draft:** Fumika Mi-ichi, Hiroshi Tsugawa.

**Writing – review & editing:** Fumika Mi-ichi, Hiroshi Tsugawa, Hiroki Yoshida, Makoto Arita.

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
