## [Decision Letter · Decision Letter 0]

20 May 2024

Dear Dr. Mi-ichi,

Thank you very much for submitting your manuscript "Characterization of Entamoeba fatty acid elongases; validation as targets and provision of promising leads for new drugs against amebiasis" for consideration at PLOS Pathogens. As with all papers reviewed by the journal, your manuscript was reviewed by members of the editorial board and by several independent reviewers. In light of the reviews (below this email), we would like to invite the resubmission of a significantly-revised version that takes into account the reviewers' comments.

The reviewers have clearly expressed their concerns that should be addressed so they will not be reiterated here. We cannot make any decision about publication until we have seen the revised manuscript and your response to the reviewers' comments. Your revised manuscript is also likely to be sent to reviewers for further evaluation.

Sincerely,

William A. Petri, Jr.

Academic Editor

PLOS Pathogens

Dominique Soldati-Favre

Section Editor

PLOS Pathogens

Michael Malim

Editor-in-Chief

PLOS Pathogens

orcid.org/0000-0002-7699-2064

Reviewer's Responses to Questions

**Part I - Summary**

Reviewer #1: This paper partially characterizes five fatty acid elongases (FAE) of Entamoeba histolytica, cause of amebic dysentery and liver abscess, as well as Entamoeba invadens, a reptilian parasite that is a model for cyst wall formation in vitro. Entamoeba FAE are of interest because 1) the protist makes very long fatty acids, 2) there are five copies of FAE in Entamoeba rather than one copy in other protists, and 3) FAE, which are present in plants but are absent in the human host, are targets for herbicides.

The problems with the paper begin with the introduction, which states with two inadequate references that Entamoeba is a "global public health problem with inadequate clinical options and high morbidity and mortality rates." Compared to what? The overwhelming evidence is that Giardia and Cryptosporidium are much more difficult to treat and cause much more morbidity and mortality than Entamoeba.

Figs. 1B and 1C, which make the simple point that Entamoebae can use 13C-oleic acid to make long chain fatty acid that are not desaturated, can be consolidated into a single plot without non-labeled cells.

The phylogenetic tree in Fig. 2A for no reason combines two unrelated enzymes: FAE and elongases (ELO). As ELOs are absent in Entamoebae, it can be left out. Further, their complex phylogeny in 2B is not justified by the data. It is just as likely that FAE were present in the common ancestor of Amoebazoa and lost by Acanthamoeba and Planoprotostelium than acquired later in evolution, as marked (see also Fig. S2).

Overexpression data in Fig. 3 is not very persuasive with regards to function of each FAE. Do red bars represent statistical significance? If so, they should be marked as such. If not, what do they mean? Why do they not use knockdown mediated by transcriptional gene silencing via antisense small RNA, a method they used previously to study synthesis of ceramides?

By far the best experiment is Fig. 4 in which free fatty acids and very long chain ceramides of Entamoeba are knocked down by three inhibitors of plant FAE. Further, cysts become permeable to Evans blue, as they have previously shown with myriocin, an inhibitor of ceramide synthesis.

In summary, this paper extends previous studies of ceramide synthesis, which had more discoveries and better methods but was published in a less prestigious journal.

Reviewer #2: In their research, Mi-ichi et al. employed lipidomics to investigate the fatty acid elongation process in Entamoeba histolytica, a protozoan responsible for amebiasis. Previous studies by the authors highlighted the presence of very long and/or medium acyl chains in this parasite, associated with glycerophospholipids and sphingolipids. Additionally, the authors characterized E. histolytica fatty acid elongases (EhFAEs), typically found in the fatty acid elongation cycle of photosynthetic organisms, despite Entamoeba being a non-photosynthetic protist. As part of their investigative approach, the researchers opted to overexpress each of the five identified EhFAEs in the parasite, tagging them with HA proteins, followed by lipidomic analysis. This enabled them to discern alterations in fatty acid profiles and subsequently infer the fatty acid elongation pathway in E. histolytica. Notably, the presence of plant-type FAE genes in E. histolytica prompted the authors to evaluate the efficacy of plant VLCFA biosynthesis inhibitors against the parasite.

Their findings revealed that these inhibitors impede the proliferation and encystation of the parasite, as demonstrated using the E. invadens model. Importantly, these compounds did not exhibit toxicity towards human foreskin fibroblasts. The authors also shed light on the mechanism behind encystation inhibition, which involves heightened membrane permeability.

This study offers valuable new insights into the mechanism of fatty acid elongation in this parasite, revealing the presence of plant-type FAE genes and functions, and assessing the effectiveness of plant VLCFA biosynthesis inhibitors in inhibiting parasite proliferation and encystation. While the findings are intriguing, certain key points require further attention in this manuscript

Reviewer #3: This manuscript leveraged lipidomics and genetic overexpression models to identify fatty acid elongases in E. histolytica as a plausible drug target against both trophozoite and cyst forms of the parasite. Lipids of interest were identified by overexpression of E. histolytica FAEs followed by lipidomics. These results identified roles for amebic FAEs in generating medium and very long chain fatty acids. FAEs present in E. histolytica are evolutionarily distinct from mammalian orthologues and are instead related to those found in photosynthetic protists and plants. Herbicides used for targeting very long chain FAEs in plants were shown to inhibit trophozoite proliferation and cyst formation in the low micromolar range while remaining non-toxic to a human cell line. The data presented in this manuscript makes a substantial contribution to the understanding of E. histolytica lipid metabolism and identifies a promising new drug target for trophozoites and cysts.

**Part II – Major Issues: Key Experiments Required for Acceptance**

Reviewer #1: Reject

Reviewer #2: The rationale behind overexpressing EhFAE genes is unclear. Are EhFAE genes expressed differentially during differentiation, exposure to various in vitro and in vivo models to study amebiasis, or under stress conditions, which could justify this approach? The authors possess ample transcriptomic data for this parasite that could be referenced to provide a physiological rationale for their decision to overexpress EhFAE genes. While the authors have demonstrated overexpression at the mRNA level, it's also crucial to confirm overexpression at the protein level through Western blotting and using HA antibodies. The authors should provide more detailed information in the Materials and Methods section regarding the concentration of G418 used to grow their transfected parasites. Specifically, it is important to clarify whether the same concentration was applied to all transfectants. This detail is crucial because the fold change in EhFAE expression varies significantly depending on the EhFAE species being overexpressed.

It remains unclear why the authors did not opt for a loss-of-function approach, such as RNAi-based gene silencing or other available methods for this parasite.

Another notable aspect of the methodology of this manuscript is the complete absence of biochemical characterization of the various EhFAE proteins identified in this study. Furthermore, such characterization would be helpful in elucidating the mode of action of the herbicide compounds utilized in this study.

The data concerning the use of herbicides to hinder the growth and differentiation of Entamoeba are intriguing. A proposed mechanism of action related to Ei suggests that cyst permeability is impaired upon herbicide treatment. However, what effects herbicide treatment may have on membrane permeability in Eh trophozoites remain to be elucidated.

A recent review has underscored the pivotal role of lipids in E. histolytica pathogenesis (doi: 10.3389/fcimb.2020.00075, certainly a valuable addition to the reference list of this manuscript). In my view, the manuscript would greatly benefit from in vivo insights into the virulence of the parasite when its EhFAE activities are impaired or at least from investigating the effects of herbicide treatments.

Reviewer #3: 1. There are no statistics indicated for Fig S1 or Fig 3. Were these data analyzed for significant differences? The findings of Fig 3A-B in particular are very central to the paper’s conclusions and would benefit from significance testing.

**Part III – Minor Issues: Editorial and Data Presentation Modifications**

Reviewer #1: not relevant

Reviewer #2: lane 28-29. It causes amebiasis for which clinical options are inadequate; therefore, therapeutic advances are urgently needed. This sentence is certainly overstated. Very efficient treatments are available for amebiasis, such as metronidazole (please mention current treatment in the introduction). Same remark for lane 62-63.

Reviewer #3: 1. Line 86 appears to have a typo. I believe the word “gnomically” is meant to be “genomically”

2. Because the available information of AmoebaDB changes with every version, it is useful for reproducibility to note in manuscript methods or main text which release number of AmoebaDB was used.

3. “Cyst” is misspelled in Table 1 as “Cyt”

2. The concentrations listed in line 242 are referred to as “~IC50s” but these values are 2-3x the IC50. Adding graphs of the full curves for IC50 experiments, would be beneficial for the reader to determine the expected inhibition at concentrations other than the IC50.

4. It would be helpful to list the criteria for determining cyst formation within the manuscript in addition to referencing the previous methods paper. For example, the authors could specify something like “Calcofluor positive, Evan’s blue negative cells were counted as mature cysts.”

5. Do VLC Cer-NDSs have known roles in affecting membrane permeability? If so, this would strengthen the paper’s arguement that VLC Cer-NDSs are the cause of encystation defects in FAE inhibitor treated samples.

PLOS authors have the option to publish the peer review history of their article (what does this mean?). If published, this will include your full peer review and any attached files.

Reviewer #1: No

Reviewer #2: No

Reviewer #3: No
---

## [Editor Report · Decision Letter 1]

19 Jul 2024

Dear Dr. Mi-ichi,

We are pleased to inform you that your manuscript 'Characterization of Entamoeba fatty acid elongases; validation as targets and provision of promising leads for new drugs against amebiasis' has been provisionally accepted for publication in PLOS Pathogens.

Best regards,

William A. Petri, Jr.

Academic Editor

PLOS Pathogens

Dominique Soldati-Favre

Section Editor

PLOS Pathogens

Michael Malim

Editor-in-Chief

PLOS Pathogens

orcid.org/0000-0002-7699-2064
---

## [Editor Report · Acceptance letter]

1 Aug 2024

Dear Dr. Mi-ichi,

We are delighted to inform you that your manuscript, " Characterization of Entamoeba fatty acid elongases; validation as targets and provision of promising leads for new drugs against amebiasis ," has been formally accepted for publication in PLOS Pathogens.

Best regards,

Michael Malim

Editor-in-Chief

PLOS Pathogens

orcid.org/0000-0002-7699-2064